# Essential Oil Biodiversity of *Achillea ligustica* All. Obtained from Mainland and Island Populations

**DOI:** 10.3390/plants11081054

**Published:** 2022-04-13

**Authors:** Ammar Bader, Aljawharah AlQathama, Pier Luigi Cioni, Lucia Ceccarini, Mohamed I. S. Abdelhady, Wajih Al-Shareef, Roberta Ascrizzi, Guido Flamini

**Affiliations:** 1Department of Pharmacognosy, Faculty of Pharmacy, Umm Al-Qura University, Makkah 21955, Saudi Arabia; aaqathama@uqu.edu.sa; 2Dipartimento di Farmacia, Università di Pisa, Via Bonanno 6, 56126 Pisa, Italy; pierluigi.cioni@libero.it (P.L.C.); guido.flamini@unipi.it (G.F.); 3Department of Agriculture, Food and Environment, University of Pisa, Via del Borghetto 80, 56124 Pisa, Italy; lucia.ceccarini@unipi.it; 4Pharmacognosy Department, Faculty of Pharmacy, Helwan University, Cairo 11795, Egypt; mohibrahem@yahoo.com; 512 Tekne’ Ricerche, Cittadella della Ricerca, S.S.7 Mesagne, 72100 Brindisi, Italy; wajihalsharif@yahoo.it

**Keywords:** Ligurian yarrow, fragranyl acetate, fragranol, GC-MS, NMR, Asteraceae, chemical polymorphism

## Abstract

Background: The genus *Achillea* is rich in essential oil (EO) with high chemical diversity. In this study, eight EO samples obtained from flowers and leaves of *Achillea ligustica* All. collected on the Mediterranean mainland and island locations were analyzed to evaluate their possible chemical diversity. Methods: Sixteen samples of EO were analyzed by GC-MS, leading to the identification of 95 compounds in the leaves and 86 compounds in the flowers; a statistical analysis was performed to determine the chemical polymorphism. Results: Monoterpenes, such as β-pinene, borneol, ɑ-terpineol and *iso*bornyl acetate, were more abundant in the continental samples, while the insular samples were richer in 1,8-cineole. Fragranyl acetate and fragranol were detected in remarkable concentrations in sample 8. The fruits of sample 8 were then cultivated under controlled agronomic conditions, providing plants rich in these compounds (sample 9). The geographical variability influenced the EO compositions, with unique observed chemotypes and a high degree of diversity among samples collected in various areas (mainland or island). Statistical analyses did not reveal any pattern between the geographical provenience and the compositions. Conclusion: Samples were distributed based on the plant organ, confirming the already reported high degree of chemical polymorphism of this species. Sample 8 could be used as a source of fragranol and fragranyl acetate, with potential applications in the insecticidal and pheromone industries.

## 1. Introduction

Essential oils are complex mixtures of volatile components, belonging to different chemical classes, produced as secondary metabolites in both normal or pathological conditions [1,2]. Essential oils are attracting increasing attention due to their importance in several applications in sectors such as aromatherapy, pharmaceuticals, cosmetics, foods, flavours and fragrances [3,4]. They have a range of pharmacological properties, such as antibacterial, antifungal, insecticidal, anti-inflammatory, antioxidant and cytotoxic activities [5,6,7,8].

Different species of *Achillea* are known for their pharmaceutical, cosmetic and aromatic properties [9,10]. *Achillea ligustica* All. (Asteraceae, Anthemideae), commonly known as Ligurian yarrow, is an aromatic plant that grows on arid slopes of the Mediterranean regions at altitudes between 0 and 800 m above sea level (a.s.l.), mainly along the Tyrrhenian coasts, from Liguria to Sicily [11]. This plant has been used in several traditional systems, such as Italian folk medicine, for skin disorders and rheumatism [9]. It is also used in Sicily against the intestinal worms [12]. In Sardinia, an infusion of *A. ligustica* is traditionally used for gastralgia and neuralgia [13]. In addition, it has traditionally been used to relieve sprains and insect bites, as well as to stop bleeding [14].

*A. ligustica* contains several secondary metabolites, such as essential oils, flavonoids and sesquiterpene lactones, piperidine amides and guaianolides, with the essential oils used as traditional herbal remedies as well as in food and cosmetics [9,10,15]. There are phytochemical studies on the essential oils of *A. ligustica* obtained from plants that grow in different locations, for example Greece [16], Northern Italy [17], Corsica [14], Sardinia [18], Sicily [19] and Central Italy [9].

It has been found that the ethanolic extracts of the flowering aerial parts of *A. ligustica* from Italy contain flavonoids and flavonoids glycosides based on luteolin, apigenin and kaempferol aglycones [20]. Other studies have reported various chemical classes, including piperidine amides, guaianolides (e.g., matricarin, chrysartemin A and B and isoapressin), novel sesquiterpene lactones with rare 5/6/5 skeletons, named ligustolide-A and ligustolide-B, 1,10-*seco*-guaianolides and a chlorine-containing sesquiterpene lactone [21,22,23].

It has been reported that several biological activities are due to essential oils or methanolic extracts of *A. ligustica*. Essential oils have been found to possess antifungal properties, as well as antibacterial activity towards both Gram-positive and -negative bacteria [18]. Moreover, anti-proliferative activities have been reported for essential oils, which could make them candidates for anti-carcinogenic formulations [9]. Both the essential oils and the methanolic extract showed anti-oxidant activity, while, for the *n*-hexane fraction, anti-diabetic properties due to α-amylase inhibition are reported [9,10].

Phytochemical studies of the essential oil (EO) of *A. ligustica* have shown a high variability in its constituents. The composition of leaf and flower EOs of *A. ligustica* from continental Greece is dominated by linalool (28.1% for leaf and 70.8% for flower) [16]. In contrast, the EOs obtained from aerial parts of specimens from northern Italy were rich in artemisia ketone (43.9%) and 2,7-dimethyl-4,6-octadien-2-ol (16.1%), and characterized by fair amounts of linalool (9.6%) [17]. Differences also emerged in the composition of EOs obtained from Mediterranean islands, with Sardinian and Corsican samples containing santolina alcohol (6.7–21.8%; 3.8–10.1%), artemisia ketone (0.3–7.6%; 3.2–7.5%) and camphor (0.7–5.8%; 17.0–17.4%) [18,24]. This high variability in the chemical composition of the EOs obtained from both the continent and the islands has an impact on their biological activity, demonstrated by different ranges of inhibition of microbial growth, which largely depend on geographical and seasonal factors [9].

Due to this variability and as a part of an ongoing study on Italian aromatic plants [25], the aim of the present study was the analysis of the essential oil composition of the leaves and flowers of *A. ligustica* from eight Italian locations. The samples were collected from three mainland locations in Southern Italy and five island locations, including Sicily and Vulcano Island in the Aeolian Archipelago; moreover, a specimen cultivated under a controlled condition was also included, in order to detect qualitative and quantitative differences with respect to geographical location and environmental conditions. GC-MS has been used to verify the influence of the different habitats on the production of volatile oils, and NMR was applied to confirm the detection of fragranyl acetate, the lead volatile compound only characterized in samples 8 and 9, with potential biological applications.

## 2. Results

### 2.1. Essential Oil (EO) Compositions

The essential oils (EOs) obtained by hydrodistillation of nine leaf and flower samples of *A. ligustica* (3 from mainland, 5 from island and 1 cultivated, Figure 1) showed degrees of variability in the content and composition of their components, with extraction yields ranging from 0.38 to 0.7% *v/w*.

A total of 95 compounds were identified in all the leaves EOs, as reported in Table 1, and 86 compounds were detected in the flowers EOs Table 2. Oxygenated monoterpenes were detected as the most abundant chemical class in all the leaf EOs, with the only exception being sample **6**, as they ranged from 37.6% (**1**) up to 70.6% (**8**). Among them, however, qualitative differences between the higher contributions were observed. Borneol was the most abundant monoterpene hydrocarbon in sample **1** (10.2%), while 1,8-cineole prevailed in samples **2** (11.8%), **3** (10.4%) and **5** (34.8%); *cis*-*p*-menth-2-en-1-ol was the most represented only in sample **4** (16.7%), while α-thujone was found only in sample **7** with a relative abundance of 21%; finally, fragranyl acetate was found only in samples **8** (54.3%) and **9** (24.0%). To the best of our knowledge, this is the first time that fragranyl acetate has been isolated from this species. The essential oils of sample **8** (Messina) showed a unique compositional profile because of the high percentages of fragranyl acetate and fragranol. These monoterpenes reached 54.3 and 8.5% in the leaves, and 59.7 and 16.9% in the flowers (Table 2), respectively. Fragranyl acetate identification was confirmed by purification and NMR analysis. The cultivated sample **9** differed, as fragranol and fragranyl acetate were markedly reduced in the leaf EO, with fragranyl acetate dropping to 24.0%, and fragranol to 3.5%. In the flower EO, instead, although the percentage of fragranol dropped from 16.9% (sample 8) to 4% (sample 9), the amount of its acetate rose from 59.7% to 71.4%. Interestingly, the sum of fragranol and fragranyl acetate in the flower oil of the wild plant (76.6%) was comparable to that of the cultivated one (75.4%). Fragranol was isolated for the first time from *Artemisia* fragrance by Bohlmann et al. in 1973 [26]. Interestingly, fragranol, which has a cyclobutane ring, is the diastereomers of grandisol (Figure 2), the main component of the aggregation pheromone, known as grandlure, of the male cotton ball weevil (*Anthonomus grandis*), which strongly attracts both males and females. It is also the main constituent of the aggregation or sex pheromones of other serious agricultural pests, such as bark weevils and bark beetles that cause conifer infestations in North America and Central Europe [27]. Traps filled with grandisol are used as a protective measure against heavy damage to cotton crops and its associated economic consequences [28]. Fragranol was found to be 100–200 times less active as an attractant than grandisol [29].

Oxygenated monoterpenes were also the most abundant chemical group in all flower samples, exhibiting relative abundances ranging from 58.4% (sample **2**) to 84.1% (sample **9**). Samples **1**, **3**, **4**, **7**, **8** and **9** contained the same oxygenated monoterpenes found in the corresponding leaf EOs as the most abundant in their compositions. Linalool was the most represented compound in samples **2** (23.1%) and **5** (26.0%), but its relative abundance was quantitatively relevant in all flower samples, while its contribution in leaf samples was always lower than 1.0%. The higher relative quantity of linalool in flower EOs is consistent with its attractive power towards pollinators [30], as also reported for *Scutellaria altissima* L. [31]. Linalool was also found as the most abundant compound in *A. ligustica* EO hydrodistilled from flowers of specimens gathered in Central Italy [32]. Santolina alcohol was exclusively found (19.3%) in sample **6**.

Among the non-oxygenated monoterpenes, which represent the second most abundant chemical class in the EO samples of three leaves (**1**, **4** and **5**) and four flowers (**1**, **3**, **4** and **7**), β-pinene was detected as the main constituent in most of the leaf (**1**, where it was also the most abundant compound in the total composition, **2**, **3**, **4**, **5** and **9**) and flower (**1**–**6**) samples. Its relative abundance was higher in the three mainland leaf samples (**1**–**3**) than in those of the islands (**4**–**8**). Sabinene, *p*-cymene and γ-terpinene followed among the quantitatively relevant compounds of this group.

### 2.2. Statistical Analysis

Due to the high dimensionality of the resulting data (two 9 × 94 and 9 × 85 matrices for leaves and flowers, respectively), multivariate statistical analysis was used to highlight possible similarities between the different samples. In the hierarchical cluster analysis (HCA), all the EO samples were distributed in two macro-clusters, as shown in Figure 3. The first macro-cluster was further divided in three sub-groups (red, green and blue), while the second macro-cluster was composed only of yellow samples. Within each sub-cluster, samples **6**, **7** and **8** showed a high degree of similarity between the compositions of the EO of leaves and flowers, as shown by their proximity in the same sub-clusters (green for sample **7**, blue for sample **6** and yellow for sample **8**). The highest degree of dissimilarity between the two organs was instead highlighted for sample **9**, as the two oils were grouped not only into different subgroups, but even into two different macro-clusters. All the other samples were instead grouped in the red sub-cluster, where they were further distributed quite sharply according to the organ of origin.

## 3. Discussion

For both organs, leaf and flower, oxygenated sesquiterpenes have always prevailed over hydrocarbons in essential oils (Table 3). In leaf sample **6**, they were the most represented chemical class and, among them, β-eudesmol reached 30.8%, thus being the main constituent in this sample. Viridiflorol, instead, was the most abundant in all other leaf and flower EO samples.

Within sesquiterpene hydrocarbons, germacrene D was found to be the main one in leaf samples **1**, and **4**–**9**, and in flower samples **3**–**6**, with an overall higher presence in the island samples. β-Caryophyllene was, instead, the most represented in both leaf and flower EOs of sample **2**, as well as in flower samples **7**–**9**. Both germacrene D and β-caryophyllene biosynthesis are reported in the literature as herbivory-induced [33]; this is consistent with our findings, as both these compounds were identified in higher relative percentages in the leaves than in flowers EOs. Finally, *allo*aromadendrene was found as the sesquiterpene hydrocarbon with the highest relative abundance in leaf sample **3**, while δ-cadinene in flower sample **1**.

All the EOs showed a very varied chemotype distribution, in agreement with previous published studies [18]. Moreover, this marked variability of the EO composition among specimens belonging to the same species has also been reported for other *Achillea* species, such as *A. millefolium* [34], *A. cartilaginea* [35], *A. biebersteinii* [36], *A. crithmifolia* [37], *A. ageratum* [33] and *A. wilhelmsii* [34]. This chemical polymorphism, thus, appears to be a distinctive trait of this genus, whose EO profile seems to be strongly influenced by the growing environment.

The score and loading plots of the principal component analysis (PCA) are shown in Figure 4 (left and right, respectively). Only the EOs from both organs of samples 8 and 9 are plotted (Figure 4, left) in the right quadrants (PC1 > 0), due to their fragranyl acetate and fragranol content (Figure 4, right). The high degree of similarity between the compositions of the EO of leaves and flowers for samples **6**, **7** and **8,** demonstrated in the HCA dendrogram, is confirmed by the score plot of the PCA (Figure 4, left), as they are grouped quite close to each other. Moreover, sample **7** is grouped closer to all the red samples, mostly in the second quadrant (PC1 > 0, PC2 > 0) of the PCA, where its positioning was mainly determined by the α-thujone vector (Figure 4, right): this compound was only detected in this sample, and with high relative concentrations in both its organs. The flower and leaf EOs of sample **6** were, instead, positioned in the third quadrant (Figure 4, left) due to their β-eudesmol, camphor and santolina alcohol content (Figure 4, right). All the red samples were distributed between the second and third quadrants, quite clearly grouped according to the organ.

Both the HCA and PCA did not prove a distribution based on the provenience but rather on the hydrodistilled organ. These data further confirm the reports already published on this species regarding the high degree of chemical variability among specimens of different geographical origin [18], as well as among different species of the *Achillea* genus [33,34,35,36,37,38]. That geographic origin drastically affects the composition of secondary plant metabolites has also been observed for other species. For example, the populations of *Peumus boldus* growing in different areas of Chile have different compositions of essential oils: plants growing in the north were rich in ascaridole while the trees growing in the south were rich in *p*-cymene [39]. This phenomenon is also observed among the populations of the genus *Achillea* growing in the same country; we can cite *Achillea millefolium* growing in different European countries [40], and *Achillea wilhelmsii* growing in Iran and Turkey. The Iranian sample was rich in caryophyllene oxide (12.5%) and *cis*-Nerolidol (10.8%), while the Turkish sample contains mainly camphor (46.6%) [41,42].

It has been shown that plants containing fragranol and fragranyl acetate are active against several species of parasites, i.e., the essential oil of *Achillea santolina* growing in Egypt, rich in fragranyl acetate and fragranol (27.3 and 8.2%, respectively), resulted in being active against the Khapra beetle (*Trogoderma granarium*) in topical application, contact and fumigation bioassays [43]. The EO of *Achillea umbellata* growing in Serbia is also rich in fragranyl acetate and fragranol (44.7 and 29.9%, respectively), and has been shown to have anxiolytic, antinociceptive and antimicrobial properties. Pure fragranyl acetate was very effective against *Staphylococcus aureus*, *Klebsiella pneumoniae* ATCC 10031 and clinical isolates, with an MIC of 0.78 µg/mL. In the same study, the microorganisms were more sensitive to fragranol, with an MIC of 0.39 µg/mL against *Staphylococcus aureus* and the clinical isolate of *Klebsiella pneumoniae*; *Klebsiella pneumoniae* ATCC 10031 was even more sensitive, with an MIC of 0.09 µg/mL [44].

## 4. Materials and Methods

### 4.1. Plant Material

The plant material was gathered at the end of April during the flowering stage in eight different sites in Southern Italy. The plants were identified by Rosalba Villari, and a voucher specimen number (E1) was deposited in the Herbarium Messanaensis, Dipartimento di Scienze Chimiche, Biologiche, Farmaceutiche ed Ambientali, Universita’ di Messina, Italy. The altitude and the pH of the soil of all sites of collection are shown in Table 4.

### 4.2. Plant Cultivation

The fruits of sample 8 (Messina, Italy) were collected in June and cultivated in a completely different habitat near Pisa, under controlled agronomic conditions (sample 9). The fruits were cultivated in a field within the Centro Sperimentale di Rottaia of Dipartimento di Scienze Agrarie, Alimentari e Agro-ambientali (University of Pisa, Italy). Soil chemical-physical properties were as follows: sand 43.8%; silt 44.9%; clay 11.3%; pH 7.9; organic matter (Lotti method) 3.0%; total N (Kjeldahl method) 1.5‰; assimilable P (Olsen method) 16.6 ppm; exchangeable K (Intern. method) 222.5 mg/kg. This soil, characterized by the presence of a rather superficial phreatic water-bearing (at most 120 cm), is especially deep and cool and has a water capacity of field (−0.033 MPa) equal to 27.3% of dry weight and a withering point (−1.5 MPa) equal to 9.4% of dry weight. The plants were transplanted in May; the inter-row and inter-plant spacing were 0.2 m. Basic fertilization consisted of 50 kg ha^−1^ of nitrogen (from urea) and 100 kg ha^−1^ of phosphorous (from triple phosphate rock) and 100 kg ha-1 of potassium (from potassium sulfate), while an additional 50 kg ha^−1^ of nitrogen (from urea) was distributed as a surface fertilizer. Weed control was accomplished by mechanical means (hoeing and harrowing) from the time of transplanting to the complete elimination of the spaces between the rows. When all of the plants were in the full flowering phase, the flowers and leaves were harvested separately.

### 4.3. Essential Oil Hydrodistillation

The bulk samples of leaves and flowers were dried separately in the shade and then 50 g amounts were submitted to hydrodistillation in a Clevenger-like apparatus for 2 h. The essential oil yields varied between 0.38 to 0.7% (dry weight).

### 4.4. Gas Chromatography with Electron Impact Mass Spectrometry (GC-EIMS)

GC-EIMS analyses were performed with a Varian CP-3800 gas-chromatograph (Varian Inc. Palo Alto, CA, USA)equipped with a DB-5 capillary column (30 m × 0.25 mm; coating thickness 0.25 μm) and a Varian Saturn 2000 ion trap mass detector. Analytical conditions were as reported in previous works [45,46]. Identification of the constituents was based on the comparison of their retention times with those of authentic samples, by comparing their l.r.i.s to the series of *n*-hydrocarbons, and by computer matching against the commercial National Institute of Standards and Technology 2014 [47] and a home-made library mass spectrum, built up from pure compounds, components of essential oils of known composition and MS literature data [48].

### 4.5. Compound Isolation

Fragranyl acetate (60 mg) was obtained in pure form from the essential oil (400 mg) of sample 8 (from Messina) by silica gel chromatography eluting with *n*-hexane (100 mL), *n*-hexane-Et_2_O 98:2 (100 mL) and *n*-hexane-Et_2_O (95:5). The latter fraction contained the pure compound fragranyl acetate.

### 4.6. Nuclear Magnetic Resonance Spectroscopy (NMR) of Fragranyl Acetate

^1^H-, ^13^C-NMR and DEPT spectra were obtained with a Bruker Avance 400 spectrometer (Bruker Corporation, Milano, Italy) in CDCl_3_, using TMS as the internal standard.

^1^H NMR (400 MHz, CDCl_3_) δ: 0.87 (3H, s), 1.35 (1H, m), 1.58 (3H, s), 1.69 (H, brs), 1.71–1.88 (4H, m), 1.93 (1H, m), 1.98 (3H, s), 2.54 (1H, m), 4.04 (2H, m), 4.55 (1H, brs), 4.78 (1H, m);

^13^C NMR (100 MHz, CDCl_3_) δ: 19.1 (CH_3_), 19.6 (CH_2_), 21.0 (CH_3_), 22.9 (CH_3_), 29.9 (CH_2_), 40.7 (C), 41.9 (CH2), 50.3 (C), 61.7 (CH_2_), 109.9 (CH_2_), 145.2 (C), 171.1 (C).

### 4.7. Statistical Analysis

The hierarchical cluster (HC) and principal component (PC) analyses were performed on the complete EO compositions of all samples. The data comprising all the detected compounds in all samples comprised an 18 × 110 matrix (18 samples, 110 compounds). The HCA was conducted with the Ward’s algorithm on unscaled data, using Euclidean distances as a measure of similarity. To perform the PCA, linear regressions were operated on mean-centered, unscaled data of the covariance matrix, to select the two highest principal components (PCs). This unsupervised method reduced the dimensionality of the multivariate data of the matrix, while preserving most of the variance. The chosen PC1 and PC2 studied 60.70% and 8.79% of the variance, respectively, for a total of 69.49% observed data. The observation of the groups of samples with the HCA and the PCA methods can be applied even without reference samples to be used as a training set to establish the model.

## 5. Conclusions

The EO chemotypes exhibited a highly variable pattern, with no correlation with the area of origin. Indeed, samples showed a remarkable degree of variability according to each sampling site, regardless of mainland or island origin. The high variability of the essential oil composition between samples is a commonly reported trait for other species of the *Achillea* genus.

Since chemotypes are linked to genotype variations, these results suggest that the investigated samples are genetically different and deserve further biomolecular analyses to confirm that any genetic differences were actually responsible for this variability. Finally, such chemical diversity of essential oil can be exploited by further investigating these samples and isolating larger quantities of secondary metabolites with potential therapeutic and biological effects, such as fragranol and fragranyl acetate.

## Figures and Tables

**Figure 1 plants-11-01054-f001:**
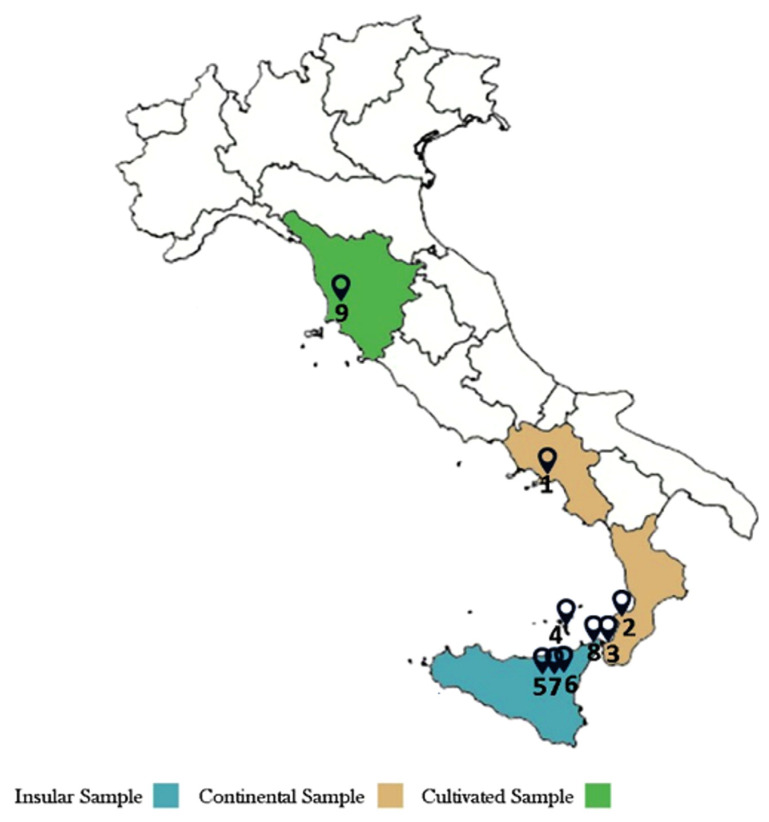
Map of samples (1–9) distributions.

**Figure 2 plants-11-01054-f002:**
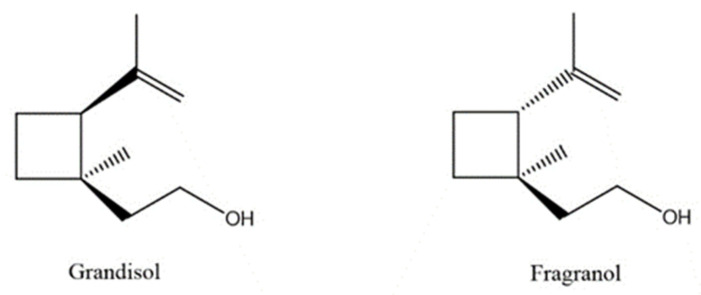
Fragranol and its diastereomers grandisol.

**Figure 3 plants-11-01054-f003:**
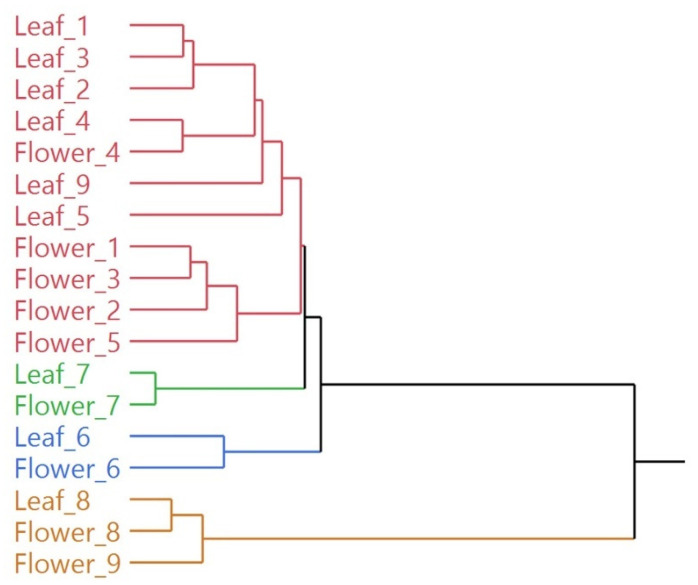
Dendrogram of the hierarchical cluster analysis (HCA) of the complete EO compositions for all samples.

**Figure 4 plants-11-01054-f004:**
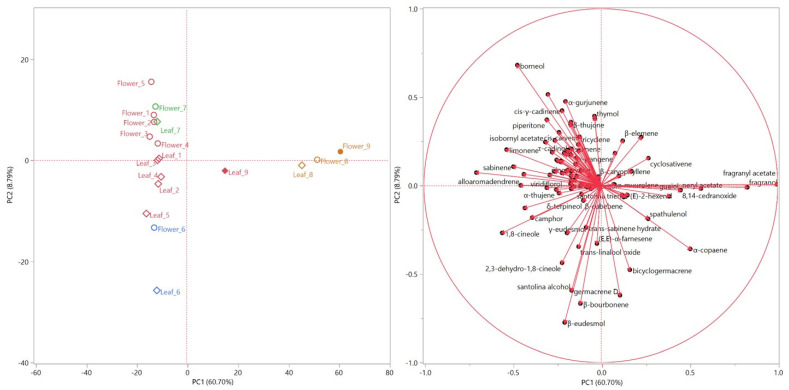
Score (**left**) and loadings (**right**) plots obtained with the Principal Component Analysis (PCA) performed on the complete compositions of all the hydrodistilled essential oils.

**Table 1 plants-11-01054-t001:** Compositions of the leaf EOs obtained from the eight wild samples of *Achillea ligustica* (1–8) and a cultivated one (9).

Compound	l.r.i ^a^	1	2	3	4	5	6	7	8	9
(*E*)-2-Hexenal	845	-	-	-	-	-	-	-	-	0.1
1-Nonene	892	-	-	-	tr	-	-	-	-	0.1
Santolina triene	910	-	-	-	-	-	-	-	-	0.3
α-Thujene	933	0.3	0.2	-	0.1	0.2	-	0.2	0.1	0.2
α-Pinene	940	2.5	0.9	0.8	0.6	2.7	0.2	1.2	0.2	0.9
Camphene	955	3.7	0.1	0.5	0.2	0.6	-	0.7	-	0.4
Benzaldehyde	963	-	-	-	-	-	-	-	-	0.1
Sabinene	978	3.4	**5.7**	0.9	1.8	4.6	1.3	3.4	0.2	0.7
β-Pinene	981	**12.0**	**8.4**	**6.4**	1.9	**17.1**	0.8	1.9	-	2.3
2,3-Dehydro-1,8-cineole	993	0.2	0.2	0.2	0.2	0.4	0.1	-	0.2	0.3
Yomogi alcohol	998	-	-	-	-	-	-	-	0.2	0.3
α-Phellandrene	1007	-	-	-	0.3	-	-	-	-	-
α-Terpinene	1020	0.4	0.4	0.2	0.7	0.3	0.2	0.7	0.5	1.5
*p*-Cymene	1028	0.9	1.3	0.5	0.7	1.7	0.5	1.1	0.7	2.1
Limonene	1033	0.6	0.4	0.2	0.3	0.7	-	0.3	-	0.2
Santolina alcohol	1037	-	-	-	-	-	**8.3**	-	-	-
1,8-Cineole	1039	4.6	**11.8**	**10.4**	**8.7**	**34.8**	**8.5**	**5.7**	0.1	1.6
γ-Terpinene	1064	0.8	1.1	0.5	1.1	0.9	0.5	1.3	0.9	2.7
*cis*-Sabinene hydrate	1070	4.1	**5.8**	**10.0**	**9.7**	2.1	3.6	**5.6**	2.5	**11.1**
Terpinolene	1089	0.2	0.2	0.2	0.2	0.1	0.1	0.4	0.2	0.6
Linalool	1101	0.1	0.2	0.4	tr	tr	0.7	-	tr	-
*trans*-Sabinene hydrate	1102	1.1	3.0	2.2	2.1	0.8	1.3	1.6	0.8	3.0
α-Thujone	1105	-	-	-	-	-	-	**21.0**	-	-
β-Thujone	1116	-	-	-	-	-	-	3.0	-	-
*cis*-*p*-Menth-2-en-1-ol	1123	0.2	0.2	2.9	**16.7**	tr	0.1	0.4	0.2	1.1
*trans*-Pinocarveol	1141	-	3.6	-	-	-	0.3	-	-	0.3
*trans*-*p*-Menth-2-en-1-ol	1147	0.4	-	1.7	**9.6**	-	-	0.4	0.1	0.4
Camphor	1148	0.2	0.7	1.4	1.1	0.2	**9.4**	**9.3**	-	2.8
Pinocarvone	1165	1.2	1.7	4.9	0.1	0.2	0.5	0.5	-	0.5
δ-Terpineol	1166	-	-	-	0.1	-	-	-	-	-
Borneol	1167	**10.2**	2.0	7.7	0.2	2.2	0.3	**12.1**	0.1	0.2
Isopinocamphone	1175	-	-	-	-	-	-	-	-	0.1
4-Terpineol	1182	1.4	3.1	1.8	3.0	0.8	0.8	3.2	2.1	**7.5**
Dill ether	1186	-	-	-	0.2	-	-	-	-	-
α-Terpineol	1192	3.2	**6.2**	**5.5**	**5.3**	4.6	2.3	1.6	0.1	1.0
*trans*-Piperitol	1207	0.2	-	1.0	**6.3**	-	-	0.3	0.1	0.2
Fragranol	1217	-	-	-	-	-	-	-	**8.5**	3.6
*trans*-Carveol	1219	-	tr	-	-	-	0.1	0.5	-	-
Piperitone	1254	0.8	-	-	0.4	-	-	1.0	-	-
*cis*-Chrysanthenyl acetate	1265	-	-	0.9	-	-	-	-	-	-
Isobornyl acetate	1285	**9.4**	1.2	**5.2**	0.7	tr	-	0.3	-	-
Lavandulyl acetate	1389	-	-	-	-	-	-	-	-	0.5
Thymol	1292	0.3	0.2	0.2	0.1	0.4	0.2	0.2	0.3	0.3
*trans*-Pinocarvyl acetate	1297	-	1.0	0.2	-	-	-	0.6	-	-
Carvacrol	1300	-	-	-	-	-	-	-	0.4	-
*iso*-Dihydrocarveol acetate	1297	-	-	-	-	-	-	-	0.7	-
*trans*-Carvyl acetate	1337	-	-	0.4	-	-	-	-	-	-
Fragranyl acetate	1346	-	-	-	-	-	-	-	**54.3**	**24.0**
α-Longipinene	1351	0.1	-	-	-	-	-	-	-	-
Eugenol	1357	tr	tr	-	-	-	-	-	-	0.1
*cis*-Carvyl acetate	1362	-	-	0.5	-	-	-	-	-	-
Neryl acetate	1365	-	-	-	-	-	-	-	0.1	-
Cyclosativene	1370	0.3	tr	-	-	0.1	0.1	-	0.2	0.1
α-Copaene	1377	0.2	0.2	0.2	-	0.3	0.3	-	0.3	0.3
*trans*-Myrtanol acetate	1381	-	-	0.2	-	-	-	-	-	-
Geranyl acetate	1383	-	-	3.3	-	-	-	-	-	-
β-Bourbonene	1384	-	-	-	-	-	0.1	-	-	-
β-Cubebene	1390	-	-	-	0.1	-	-	-	-	-
β-Elemene	1391	-	-	-	-	-	-	-	0.1	-
Methyl eugenol	1402	-	-	0.2	-	-	-	-	-	-
α-Gurjunene	1409	-	-	-	-	tr	-	-	-	-
β-Caryophyllene	1420	1.6	3.7	0.2	0.1	0.4	0.4	1.1	0.7	1.1
α-Himachalene	1449	0.4	-	0.2	-	-	-	-	-	-
α-Humulene	1456	0.1	0.3	-	-	tr	-	-	0.1	-
Alloaromadendrene	1461	0.3	0.4	0.3	0.1	0.8	0.7	0.3	0.2	0.2
β-Chamigrene	1475	0.2	-	-	-	-	-	-	-	-
γ-Muurolene	1475	-	0.2	-	0.1	-	-	-	0.1	-
Germacrene D	1482	1.3	0.9	0.2	1.8	1.0	**9.5**	1.6	**5.7**	1.5
β-Selinene	1493	1.0	-	-	-	-	-	0.5	-	-
Bicyclogermacrene	1496	0.2	0.1	0.2	0.2	-	0.8	0.2	0.7	0.2
α-Muurolene	1499	-	-	-	-	-	0.1	-	-	-
β-Himachalene	1499	0.2	-	-	-	-	-	-	-	-
(*E,E*)-α-Farnesene	1508	-	0.1	-	0.3	-	0.1	-	0.1	-
*cis*-γ-Cadinene	1511	0.2	0.1	-	-	-	-	-	0.1	0.1
δ-Cadinene	1525	0.7	0.3	-	0.1	tr	0.2	0.2	0.1	-
8,14-Cedranoxide	1542	-	-	-	-	-	-	-	0.5	-
α-Elemol	1549	-	-	1.1	-	-	-	-	-	0.4
*trans*-Nerolidol	1564	0.5	**6.4**	0.4	-	-	-	-	0.5	3.0
Spathulenol	1578	0.4	0.4	0.2	0.1	0.1	0.4	0.3	0.4	1.0
Caryophyllene oxide	1583	2.2	2.4	0.8	-	-	0.1	1.5	1.0	1.4
Viridiflorol	1592	3.3	**6.3**	**7.0**	2.4	**10.5**	**9.1**	**5.3**	3.7	3.8
Guaiol	1597	-	-	-	-	-	-	-	1.1	-
Cedrol	1598	-	-	-	0.3	-	-	-	-	-
1-*epi*-Cubenol	1629	-	-	-	1.5	-	-	-	-	-
γ-Eudesmol	1632	3.5	-	1.6	-	0.1	1.6	-	-	-
Caryophylla-4(14),8(15)-dien-5-α-ol	1641	-	-	-	-	-	-	-	0.5	-
T-Cadinol	1642	1.2	0.5	0.7	-	-	-	-	-	-
T-Muurolol	1643	-	-	-	0.3	-	-	-	-	-
β-Eudesmol	1651	3.5	2.8	-	-	2.4	**30.8**	-	-	0.9
α-Eudesmol	1654	-	-	0.2	-	-	-	-	-	0.8
Kongol	1655	-	**5.4**	-	1.7	-	-	**5.6**	-	-
α-Santalol	1680	-	-	-	0.3	-	-	-	-	-
α-Bisabolol	1685	-	1.6	-	-	2.8	-	-	-	-
Chamazulene	1727	0.4	0.1	0.3	0.1	-	-	-	0.1	-
Monoterpene hydrocarbons		24.8	18.7	10.2	7.9	28.9	3.6	11.2	2.8	11.9
Oxygenated monoterpenes		37.6	40.9	61.0	64.5	46.5	37.5	67.3	70.6	58.5
Sesquiterpene hydrocarbons		6.8	7.2	1.3	2.8	2.6	12.3	4.0	8.4	3.4
Oxygenated sesquiterpenes		14.6	25.8	12.0	6.6	15.9	42.0	12.7	7.9	11.3
Phenylpropanoids		tr	tr	0.2	-	-	-	-	-	0.1
*Nor*-terpenes		0.4	0.1	0.3	0.1	-	-	-	0.1	-
Non-terpene derivatives		-	-	-	0.1	-	-	-	-	0.3
Total identified		84.25%	92.7%	85%	81.9%	93.9%	95.4%	95.2%	89.9%	80.5%

^a^ Linear retention index (HP- column), tr < 0.1%.

**Table 2 plants-11-01054-t002:** Compositions of the flowers EOs obtained from the eight wild samples of *Achillea ligustica* (1–8) and a cultivated one (9).

Compound	l.r.i ^a^	1	2	3	4	5	6	7	8	9
Tricyclene	928	0.4	-	-	-	-	-	-	-	-
α-Thujene	933	-	0.1	-	0.1	-	0.1	0.2	-	-
α-Pinene	940	2.6	0.3	0.9	0.6	0.1	0.4	1.2	0.1	-
Camphene	955	**4.9**	0.1	0.5	0.4	0.2	0.1	0.7	-	-
Benzaldehyde	963	0.1	-	0.2	-	0.1	0.1	0.1	0.2	tr
Sabinene	978	4.0	2.4	0.8	1.2	0.2	0.4	**6.0**	0.1	-
β-Pinene	981	**6.7**	2.7	**5.8**	2.5	0.9	0.9	1.6	tr	tr
Myrcene	992	0.1	-	-	-	-	-	0.2	-	-
2,3-Dehydro-1,8-cineole	993	-	-	0.4	-	-	0.3	0.1	-	-
α-Phellandrene	1007	-	-	-	0.5	-	-	-	-	-
α-Terpinene	1020	0.4	0.3	0.1	0.9	0.1	0.3	0.3	0.1	-
*p*-Cymene	1028	0.9	0.4	0.5	1.1	0.5	0.1	0.7	0.7	0.2
Limonene	1033	0.7	0.3	0.2	0.3	0.2	-	0.2	-	-
Santolina alcohol	1037	-	-	-	-	-	**19.3**	-	-	-
1,8-Cineole	1038	**8.2**	**10.2**	**21.0**	**8.1**	**5.2**	**11.9**	**6.0**	0.2	0.1
γ-Terpinene	1064	1.3	0.7	0.3	1.3	0.5	0.6	0.9	0.3	0.2
*cis*-Sabinene hydrate	1070	2.0	3.3	2.5	**5.0**	1.7	0.8	4.2	1.6	0.3
Terpinolene	1089	0.2	0.2	-	0.2	tr	0.1	0.2	0.1	-
*trans*-Linalool oxide	1090	-	-	-	-	-	0.2	-	-	-
Linalool	1101	**10.7**	**23.1**	**11.4**	**13.1**	**26.0**	**8.0**	3.8	0.7	3.0
*trans*-Sabinene hydrate	1102	-	0.9	2.2	1.7	-	0.3	-	1.0	0.6
Isoamylisovalerate	1103	0.4	0.4	0.7	-	-	-	-	-	-
α-Thujone	1105	-	-	-	-	-	-	**22.6**	-	-
β-Thujone	1116	-	-	-	-	-	tr	3.3	-	-
*cis*-*p*-Menth-2-en-1-ol	1123	0.1	0.1	-	**14.4**	0.1	-	0.1	0.1	0.2
α-Campholenal	1127	-	0.1	-	-	-	-	-	-	-
*trans*-Pinocarveol	1141	0.2	2.3	1.0	-	0.2	0.4	-	-	0.4
*cis*-Sabinol	1145	-	-	-	-	-	-	1.0	-	-
*trans*-*p*-Menth-2-en-1-ol	1147	-	-	-	**9.5**	-	-	0.2	-	-
Camphor	1148	**8.2**	1.2	2.5	2.0	2.9	**13.5**	**8.8**	-	0.2
Pinocarvone	1165	0.7	-	4.0	0.1	0.2	-	0.5	-	-
Borneol	1167	**16.5**	**5.7**	**12.7**	1.1	**16.3**	1.3	**13.9**	0.1	0.4
Terpinen-4-ol	1182	1.6	2.1	1.0	3.4	1.3	2.0	1.5	0.7	2.4
α-Thujenal	1184	-	0.1	-	-	-	-	-	-	-
Dill ether	1186	-	0.2	-	-	-	-	-	-	-
α-Terpineol	1192	2.9	**5.6**	3.8	4.5	4.4	1.7	1.8	0.1	0.4
*trans*-Piperitol	1207	-	-	-	**8.0**	tr	tr	0.3	-	-
Fragranol	1217	-	-	-	-	-	-	-	**16.9**	4.0
*trans*-Carveol	1219	-	-	-	-	tr	0.3	0.7	-	-
*cis*-Carveol	1231	-	-	-	-	-	-	0.1	-	-
Cuminaldehyde	1242	-	-	-	0.1	-	-	-	-	-
Carvone	1244	-	-	-	-	tr	-	1.0	-	-
Piperitone	1254	0.5	-	-	0.4	-	-	0.8	-	-
*cis*-Chrysanthenyl acetate	1265	-	-	1.3	-	-	0.1	-	-	-
Isobornyl acetate	1285	**7.2**	2.3	**5.7**	1.8	0.2	0.2	0.3	0.1	0.4
Thymol	1292	0.9	0.2	0.3	0.1	4.1	0.7	0.1	1.1	0.3
*trans*-Pinocarvyl acetate	1297	-	0.9	0.2	-	-	-	tr	-	-
Carvacrol	1300	-	-	0.5	-	-	-	-	-	-
Myrtenyl acetate	1327	-	0.1	0.2	-	-	-	tr	-	-
*trans*-Carvyl acetate	1337	-	-	0.2	-	-	-	-	-	-
Fragranyl acetate	1346	-	-	-	-	-	-	-	**59.7**	**71.4**
α-Longipinene	1351	0.1	-	-	-	-	-	-	-	-
Eugenol	1357	-	-	-	-	tr	-	tr	-	-
*cis*-Carvyl acetate	1362	-	-	0.5	-	-	-	-	-	-
Cyclosativene	1370	0.3	0.1	0.5	-	0.1	0.1	0.1	0.3	0.2
α-Ylangene	1372	-	-	0.5	-	-	-	-	-	-
α-Copaene	1377	0.1	0.1	0.2	tr	0.2	0.1	0.1	0.4	0.2
β-Elemene	1391	-	-	-	-	0.1	-	-	-	-
α-Gurjunene	1409	-	-	-	-	0.1	tr	0.1	-	-
β-Caryophyllene	1420	0.6	0.9	0.2	-	0.1	-	1.7	1.0	0.4
α-Himachalene	1449	0.2	-	0.1	-	-	-	-	-	-
(*E*)-β-Farnesene	1455	-	0.1	-	-	-	-	0.2	-	-
α-Humulene	1456	0.1	0.1	-	-	-	-	0.2	-	-
Alloaromadendrene	1461	0.3	0.7	0.6	0.3	1.4	0.9	0.5	0.4	-
β-Chamigrene	1475	-	-	-	-	-	-	0.2	-	-
γ-Muurolene	1475	-	0.1	-	0.1	0.3	-	0.1	0.1	-
Germacrene D	1482	-	0.4	0.2	-	0.2	-	0.9	0.8	-
*ar*-Curcumene	1483	-	-	-	0.1	-	-	-	-	-
β-Selinene	1493	-	0.5	-	-	-	-	-	-	-
α-Zingiberene	1494	-	-	-	0.2	-	-	-	-	-
Bicyclogermacrene	1496	-	0.1	-	0.1	0.1	-	0.1	0.1	-
α-Muurolene	1499	-	-	-	-	0.1	-	0.1	0.1	-
*cis*-γ-Cadinene	1511	0.3	0.2	0.3	-	-	-	0.6	-	-
δ-Cadinene	1525	1.3	0.2	-	0.1	0.1	-	-	0.1	-
*trans*-Nerolidol	1564	1.1	4.4	-	-	-	-	0.1	-	1.2
Spathulenol	1578	-	0.4	-	-	0.2	0.1	0.1	0.4	0.1
Caryophyllene oxide	1583	0.5	1.3	0.3	-	0.2	-	0.7	0.3	1.0
Viridiflorol	1592	1.7	**6.4**	**5.7**	**5.1**	**22.4**	**12.7**	**4.8**	**6.0**	2.4
Caryophylla-4(14),8(15)-dien-5-α-ol	1641	-	-	-	-	-	-	-	-	0.1
T-Cadinol	1642	0.1	1.8	0.7	-	-	-	-	-	-
Cubenol	1644	1.3	-	-	-	-	-	-	-	-
β-Eudesmol	1651	-	2.9	-	-	1.2	11.4	-	-	-
Kongol	1655	-	-	-	1.7	-	-	2.2	-	-
α-Bisabolol	1685	-	0.4	-	-	3.0	-	-	-	-
Chamazulene	1727	-	0.1	tr	tr	0.1	0.1	0.1	-	-
Monoterpene hydrocarbons		22.2	7.5	9.1	9.1	2.7	3.0	12.2	1.4	0.4
Oxygenated monoterpenes		60.0	58.4	71.4	73.3	62.6	61.0	71.1	82.3	84.1
Sesquiterpene hydrocarbons		3.3	3.5	2.6	0.9	2.8	1.1	4.9	3.3	0.8
Oxygenated sesquiterpenes		4.7	17.9	6.7	6.8	27.0	24.2	7.9	6.7	4.8
Phenylpropanoids		tr	tr	0.2	-	tr	-	tr	-	-
*Nor*-terpenes		-	0.1	tr	tr	tr	0.1	0.1	-	-
Non-terpene derivatives		0.5	0.4	0.9	-	0.1	0.1	0.1	0.2	-
Total identified		90.7%	87.8%	90.7%	90.1%	95.3%	89.5%	96.3%	93.9%	90.1%

^a^ Linear retention index (HP-column), tr < 0.1%.

**Table 3 plants-11-01054-t003:** The structure of the most abundant terpenoids of *A. ligustica* samples.

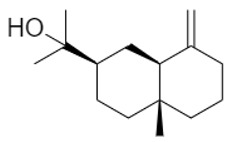	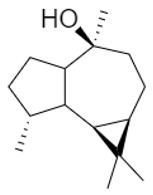	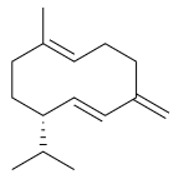
β-eudesmol	Viridiflorol	Germacrene D
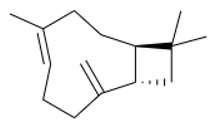	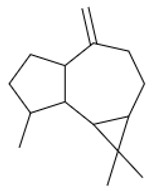	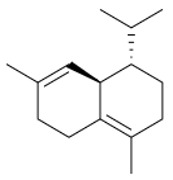
β-Caryophyllene	*allo*aromadendrene	δ-cadinene
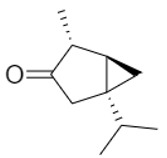	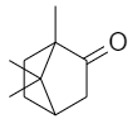	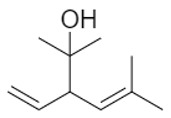
α-thujone	Camphor	Santolina alcohol

**Table 4 plants-11-01054-t004:** Altitude and soil pH of the sample collection sites.

Site of Collection	Sample Code	Origin	Altitude (m)	Soil pH
Penta di Fisciano	Sample **1**	Mainland	320	7.2
Vibo Valentia	Sample **2**	Mainland	476	6.8
Villa San Giovanni	Sample **3**	Mainland	sea level	7.2
Vulcano Island	Sample **4**	Island	sea level	5.4
Montalbano	Sample **5**	Island	908	7.2
Novara di Sicilia	Sample **6**	Island	650	6.6
Sella Mandrazzi	Sample **7**	Island	1125	6.9
Messina	Sample **8**	Island	100	7.1
Rottaia (PI)	Sample **9**	Cultivated	50	7.9

## Data Availability

Data are contained within the article.

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
