# Peer review of "Essential Oil Biodiversity of Achillea ligustica All. Obtained from Mainland and Island Populations"

_plants, 2022, doi:10.3390/plants11081054_

Round 1
Reviewer 1 Report
The manuscript entitled "Essential oil biodiversity for Achillea ligustica All. obtained 2 from mainland and island populations" is worth publishing in its current form.
Author Response
We would like to express our gratitude for the comments done by the first reviewer.
Reviewer 2 Report
The manuscript entitled "Essential oil biodiversity for Achillea ligustica All. obtained from mainland and island populations" reports a study of the essential oils of Achillea ligustica that identified 68 compounds. The authors verify the differences between insular and continental samples. The manuscript show results that can be used in further studies. There are some points to change/clarify to increase the visibility of the manuscript.
1 - Figure 1 - page 4 and page 3 line 118
"Fragranol and its enantiomer grandisol."
"fragranol, which has a cyclobutane ring, is the enantiomer of grandisol"
Please clarify if these compounds are enantiomers or diastereomers.
2 - Figure 2 and figure 3
Please, in the text the meaning of the color is explained, but to increase the visibility of the manuscript is suitable to add the captions of figures 2 and 3.
3 - Perhaps is suitable to move figure 2 after table 2.
4 - Please, a map showing the different sites of the collection will aid the interpretation of the data shown in table 3 and figures 2 and 3.
5 - Please, add a figure with the structures of the compounds cited in the discussion section.
6 - Discussion section - lines 397 - 403
"Only the EOs f... are plotted (Figure 3, left) in the right quadrants (PC1>0), due to their fragranyl acetate .... upper left quadrant (PC1>0, PC2>0) of the PCA, ..... lower left quadrant (Figure 3, left) 406 due to their β-eudesmol,.... between the upper (PC2>0) and lower (PC2<0) left (PC1<0) quadrants..."
Please, replace the terms upper, right, etc with the first quadrant, second quadrant, and so on.
7 -The criteria for using HCA and PCA are not clear, and the comparison between the two is very superficial in the discussion. Please clarify and improve the discussion.
Author Response
We would like to express our gratitude for the comments done by the second reviewer.
We have performed all the requested changes.
1 - Figure 1 - page 4 and page 3 line 118
"Fragranol and its enantiomer grandisol."
"fragranol, which has a cyclobutane ring, is the enantiomer of grandisol"
changed
Please clarify if these compounds are enantiomers or diastereomers.
They are diastereomers.
2 - Figure 2 and figure 3
Please, in the text the meaning of the color is explained, but to increase the visibility of the manuscript is suitable to add the captions of figures 2 and 3.
Done
3 - Perhaps is suitable to move figure 2 after table 2.
Done
4 - Please, a map showing the different sites of the collection will aid the interpretation of the data shown in table 3 and figures 2 and 3.
Done: Figure 1.
5 - Please, add a figure with the structures of the compounds cited in the discussion section.
Done: Table 4.
6 - Discussion section - lines 397 - 403
"Only the EOs f... are plotted (Figure 3, left) in the right quadrants (PC1>0), due to their fragranyl acetate .... upper left quadrant (PC1>0, PC2>0) of the PCA, ..... lower left quadrant (Figure 3, left) 406 due to their β-eudesmol,.... between the upper (PC2>0) and lower (PC2<0) left (PC1<0) quadrants..."
Please, replace the terms upper, right, etc with the first quadrant, second quadrant, and so on.
Done
7 -The criteria for using HCA and PCA are not clear, and the comparison between the two is very superficial in the discussion. Please clarify and improve the discussion.
Done
Reviewer 3 Report
In this interesting paper essential oil (EO) A. ligustica from eight Italian locations, consisting of eight samples of flowers and leaves of Achillea ligustica All. collected in Mediterranean mainland and island locations were analyzed by GC-MS leading to the identification of 68 compounds, a statistical analysis was performed to determine the chemical polymorphism. some Monoterpenes such as β-pinene, borneol, É‘-terpineol, and isobornyl acetate were more abundant in continental samples, while insular samples were richer in 1,8-cineole. Fragranyl acetate and fragranol were detected in remarkable concentrations in sample 8, the fruits of sample 8 were cultivated undercontrolled agronomic conditions, providing plants rich in these compounds
Claims about geographical variability influenced the EO compositions, by the authors (please add some more examples)
COMMENTS
Nuclear magnetic resonance spectroscopy (NMR) PLEASE SPECIFY compound
I would be good to add some bioactivity of isolated fraganyl acetate
Please check english language
Author Response
We would like to express our gratitude for the comments done by the third reviewer.
We have performed all the requested changes.
- Claims about geographical variability influenced the EO compositions, by the authors (please add some more examples)
Done
We added the following paragraph
That geographic origin drastically affects the composition of secondary plant metabolites has also been observed for other species. For example, the populations of Peumus boldus growing in different areas of Chile have different compositions of essential oils: plants growing in the north were rich in ascaridole while the trees growing in the south were rich in p-cymene [39]. This phenomenon is also observed among the populations of the genus Achillea growing in the same country; we can cite Achillea millefolium growing in different European countries [40], and Achillea wilhelmsii growing in Iran and Turkey. The Iranian sample was rich in caryophyllene oxide (12.5%) and cis-Nerolidol (10.8%), while the Turkish sample contains mainly camphor (46.6%) [41,42].
- Nuclear magnetic resonance spectroscopy (NMR) PLEASE SPECIFY compound
Done: fragranyl acetate
- It would be good to add some bioactivity of isolated fraganyl acetate
Done: we added the following paragraph
It has been shown that plants containing fragranol and fragranyl acetate are active against several species of parasites, i.e. the essential oil of Achillea santolina growing in Egypt, rich in fragranyl acetate and fragranol (27.3 and 8.2%, respectively), resulted active against Khapra beetle (Trogoderma granarium) in topical application, contact and fumigation bioassays [43]. The EO of Achillea umbellata growing in Serbia is also rich in fragranyl acetate and fragranol (44.7 and 29.9%, respectively), and has been shown to have anxiolytic, antinociceptive and antimicrobial properties. Pure fragranyl acetate was very effective against Staphylococcus aureus, Klebsiella pneumoniae ATCC 10031 and clinical isolates, with a MIC of 0.78 µg/ml. In the same study, the microorganisms were more sensitive to fragranol, with a MIC of 0.39 µg/ml against Staphylococcus aureus and the clinical isolate of Klebsiella pneumoniae; Klebsiella pneumoniae ATCC 10031 was even more sensitive, with a MIC of 0.09 µg/ml [44].
- Please check English language
Done